# Pluripotency-Associated microRNAs in Early Vertebrate Embryos and Stem Cells

**DOI:** 10.3390/genes14071434

**Published:** 2023-07-12

**Authors:** Pouneh Maraghechi, Maria Teresa Salinas Aponte, András Ecker, Bence Lázár, Roland Tóth, Nikolett Tokodyné Szabadi, Elen Gócza

**Affiliations:** 1Department of Animal Biotechnology, Institute of Genetics and Biotechnology, Hungarian University of Agriculture and Life Sciences; Agrobiotechnology and Precision Breeding for Food Security National Laboratory, Szent-Györgyi Albert str. 4, 2100 Gödöllő, Hungary; 2National Centre for Biodiversity and Gene Conservation, Institute for Farm Animal Gene Conservation (NBGK-HGI), Isaszegi str. 200, 2100 Gödöllő, Hungary

**Keywords:** microRNA clusters, embryonic stem cells, embryonic development, cell cycle regulators

## Abstract

MicroRNAs (miRNAs), small non-coding RNA molecules, regulate a wide range of critical biological processes, such as proliferation, cell cycle progression, differentiation, survival, and apoptosis, in many cell types. The regulatory functions of miRNAs in embryogenesis and stem cell properties have been extensively investigated since the early years of miRNA discovery. In this review, we will compare and discuss the impact of stem-cell-specific miRNA clusters on the maintenance and regulation of early embryonic development, pluripotency, and self-renewal of embryonic stem cells, particularly in vertebrates.

## 1. Introduction

microRNAs (miRNAs) are endogenous, 22–25 nucleotide long non-coding RNAs that post-transcriptionally regulate gene expression by binding to the 3′ untranslated region (UTR) of their target mRNAs and inhibiting their translation or stability. The central regulatory role of miRNAs in multiple biological processes, including cell cycle regulation, apoptosis, aging, cell fate decisions, and different signaling pathways, has been widely described. Developmental studies in vertebrates demonstrated that miRNAs are crucial molecules for embryogenesis (reviewed by [1,2]). Indeed, miRNAs play important roles in stem cell properties by regulating self-renewal and differentiation (reviewed by [3,4]). In this review, we will focus on the impact of stem-cell-specific miRNA clusters on the maintenance and regulation of early embryogenesis and embryonic stem cell (ESC) pluripotency in vertebrates, particularly in humans, mice, rabbits, and chickens.

## 2. Overview of microRNA Biogenesis

miRNA genes are transcribed by RNA Polymerase II/III into long primary transcripts (pri-miRNA), which can also be polycistronic [5,6]. A significant percentage of miRNAs are intragenic, residing within their host gene’s intronic or/and exonic regions [7]. They are transcribed from introns, and occasionally from exons of protein-coding genes, sharing the host gene’s expression regulators. Conversely, intergenic miRNAs are located between genes and transcribed independently from their own promoters [6,8]. However, some intergenic miRNAs can also be co-transcribed with their neighboring genes [9]. Many miRNAs are localized as a cluster on the same chromosome in a close proximity of maximum 10 kb [10,11]. They are co-transcribed from the same genomic loci as a single polycistronic transcript [12,13]. They also share the same seed sequence, therefore sharing common target mRNAs and biological functions [10,14]. The biogenesis of miRNA can follow canonical or non-canonical pathways.

In the canonical pathway, pri-miRNAs are mostly transcribed by RNA Polymerase II [5]. Transcribed pri-miRNA forms a hairpin structure. This structure has two domains that are recognized by the Drosha microprocessor complex (composed of Drosha and Dgcr8). Drosha cleaves the stem of the hairpin and produces a shorter hairpin, termed “precursor miRNA” (pre-miRNA) [15]. In the presence of the RanGTP cofactor, Exportin-5 translocates the pre-miRNA from the nucleus into the cytoplasm where Dicer further cleaves it and generates a short, double-stranded miRNA precursor [16]. One of the strands of the generated miRNA duplex is selected as a guide miRNA by Argonaute based on the identities of the 5′ nucleotide of each strand and the relative thermodynamic stability of the two ends of the duplex, while the other strand (miRNA*) is degraded. The guide miRNA strand is then loaded into Argonaute 2 (AGO2), building the RISC complex which is composed of the RNAse III-type nuclease Dicer and the PAZ/PIWI protein (AGO2) [17,18,19]. Once miRISC is built, it can bind to its target mRNA’s 3′-UTR through a complementary sequence. A perfect base pairing results in target degradation, while an imperfect base pairing results in translational silencing of the target mRNA. The miRNA structure contains a well-conserved sequence of six nucleotides called the “seed” that represents the most important feature of target recognition [20,21].

Non-canonical miRNAs are also derived from RNA Polymerase II; however, few miRNAs can be transcribed from RNA Polymerase III. The non-canonical miRNA biogenesis occurs through a Drosha/Dgcr8 and/or Dicer-independent manner, when miRNAs originate from introns (mirtrons) [22,23] or other small RNAs, such as snoRNAs, shRNAs, tRNAs, tRNase Z, and endo-siRNAs [24,25,26,27,28]. miRNAs derived from endo-shRNAs and tRNA precursors are transcribed by RNA Polymerase III [24,27].

In the Drosha/ Dgcr8-independent pathway (also called the mirtron pathway), introns are primarily processed in the nucleus by the spliceosome and a debranching enzyme that produces short hairpins called mirtrons. These hairpins are then exported to the cytoplasm by Exportin-5 and cleaved by Dicer [22,23]. Even though the majority of introns in vertebrates are elongated from several kilobases up to megabases in length, there are short introns of 50–150 nt in length which may form hairpin structures [29,30]. Hence, mitrons constitute only a small fraction of mammalian miRNAs. There are other subclasses of miRNAs with a potential pre-miRNA-like hairpin which are derived by an alternative Drosha/ Dgcr8-independent pathway. The processing of snoRNAs, shRNAs, tRNAs, tRNase Z, and endo-siRNA precursors to miRNAs are also independent of Drosha/ Dgcr8 [24,25,26,27,28].

In the Dicer-independent pathway, the produced pri-miRNAs are processed by Drosha into pre-miRNAs. However, these pre-miRNAs are not suitable substrates for Dicer; therefore, they are loaded into Ago2 instead, and Ago2-dependent cleavage produces an intermediate 3′ end. The final maturation occurs as the 3′-5′ trimming of the 5′ strand is completed by a cellular nuclease [31,32]. In the Dicer-independent pathway, the tRNase Z-dependent pathway precursor tRNA is cleaved by tRNase Z at the 3′ end [33].

## 3. Role of the miRNAs in Embryonic Development

Many miRNAs function during early embryonic development. The first known miRNA, lin-4, plays a crucial role in the developmental timing of stage-specific cell lineages in *C. elegans* [34]. Shortly after the discovery of lin-4, let-7 was recognized as an essential miRNA to control the pattern of temporal development of larval cell fates during the adult stage, which acts as a temporal switch between larval and adult fates. Loss of the let-7 gene activity leads to abnormalities in the pattern of larval development as a result of the reiteration of larval cell fates during the adult stage, while overexpression of the let-7 gene causes the early presence of adult cell fate during larval stages [35]. Both lin-4 and let-7 miRNAs are highly conserved; miR-125a and miR-125b are the mammalian orthologues of lin-4 [36,37].

The impact of miRNAs in embryonic development has been demonstrated in different organisms either by dissecting the miRNA pathway or by functional and expression pattern analyses of miRNAs. In mice, different miRNAs are present during all stages of embryonic development. Deletion of Dicer, Dgcr8, or Ago2 induced early post-implantation lethality [38,39,40,41]. Dicer 1 mutant mouse embryos exhibited a 50% lethality rate at the 7.5-day embryonic stage; surviving embryos exhibited morphological abnormalities and suppressed Oct4 and brachyury expression, indicating an embryonic developmental arrest and deficient pluripotency [38]. Homozygous deletion of the first and second exons of the Dicer gene resulted in severe hypomorphic embryos with retarded phenotype and mid-gestation death, suggesting an essential role for Dicer in normal mouse embryonic development [42]. Moreover, Dgcr8 knockout mouse embryos died early in development and displayed extreme morphological defects [41].

miRNA profiling uncovered developmental stage-specific miRNA expression. Zygotes inherit both maternal and paternal miRNAs [43,44]; however, the maternal miRNA pool is largely depleted between the one- and two-cell stages. Subsequently, miRNA expression is increased upon the activation of zygotic transcription [43]. Remarkably, certain stem-cell-specific miRNAs, such as miR-290 cluster members, are the first de novo expressed miRNAs in early embryogenesis [43,45,46]. Similarly, in rabbits, the expression of ocu-miR-290 cluster members begins at early embryonic stages [47]. Furthermore, ocu-miR-512-5p, which is evolutionarily related to miR-290 and its human homolog miR-371, is also expressed in early stages of rabbit embryonic development [47]. miR-29b is another important development regulator. It is highly expressed in the two-cell-stage mouse embryos and downregulated in the four-cell stage. Inhibition of miR-29b led to early developmental blockade by downregulating Dnmt3a/b and disrupting DNA methylation [48]. The placenta-specific C19MC miRNAs play significant roles in the fate of trophoblasts. For example, miR-519d regulates human trophoblast migration [49]. Additionally, a high expression of ocu-miR-512-5p, another C19MC family member, was reported in rabbit trophoblasts and hypoblasts, in line with its importance in pre-implantation [47]. Likewise, hsa-miR-512-3p miRNAs are among the most highly expressed miRNAs in human blastocysts [50]. miR-376c (member of the hsa-miR-379/miR-656 cluster) promotes cell proliferation and invasion of human trophoblast cells [51], whereas miR-155 inhibits their cell proliferation and migration [52]. The miR-302 cluster is a highly expressed stem-cell-specific miRNA cluster in rabbit pre-implantation embryos [47]. Similar to rabbits, miR-302 cluster members are highly expressed in the human blastocysts [50]. miR-21 is expressed in both human and mouse oocytes and blastocysts [53,54]. It is associated with oocyte maturation, blastocyst formation, and pre-implantation embryonic development [54]. The let-7 family also plays a crucial role in mammalian embryonic development. The expression of let-7 in mouse embryos regulates inner cell mass cell fate determination and reinforces blastocyst formation [55].

## 4. Discovery of miRNA Clusters in Stem Cells

To date, many studies have demonstrated the regulatory function of miRNAs (both canonical and non-canonical) in the self-renewal and differentiation of ESCs and iPSCs. Many canonical miRNAs, such as miR-302 and miR-290 clusters, reportedly promote ESC properties. miR-320 and miR-702 are the two first-discovered non-canonical miRNAs expressed in mouse ESCs [56]. Interestingly, miR-133a regulates the transcription of the DNMT3B gene in human ESCs through a non-canonical pathway when it is translocated to the nucleus, while it regulates skeletal and cardiac muscle function via canonical target mRNA repression in the cytoplasm [57].

DICER deletion has induced a proliferation defect, hindered teratoma and chimera formation, and constrained the differentiation of mESCs [58,59] and hESCs [60]. Similarly, Dgcr8-knockout mESCs proliferate slowly and exhibit either delayed or reduced differentiation potential, accompanied by defects in embryonic anatomy and teratoma formation [41,61]. While Dicer- and Dgcr8-mutant ESCs contributed to an overall analysis of miRNA functional aspects in ESC properties, cloning and deep sequencing of non-coding small RNAs from stem cells led to the discovery and profiling of stem-cell-specific miRNAs with potential functions in the self-renewal and differentiation of ESCs. ESC-associated transcription factors, including OCT4, NANOG, SOX2, and REX1, regulate ES-specific miRNAs through binding to their promoter region [62,63]. As a prominent example, the miR-290/295 cluster was identified as a stem-cell-specific miRNA cluster in mice by cloning and deep sequencing [64]. Subsequently, the human homolog clusters miR-302/367 and miR-371/373 were also identified by cDNA cloning in human ESCs [65]. Additionally, the deep sequencing of small RNAs showed that the rabbit ES-like cells highly express the ocu-miR-302/367 cluster, which potentially maintains pluripotency by negatively modulating Lefty [47].

### 4.1. miRNA Clusters Regulate Pluripotency

miR-302/367, miR-290/295, and their human homolog, the hsa-miR-371/373 cluster, are highly expressed in vertebrate ESCs. They modulate self-renewal and pluripotency through regulating cell cycle progression. These clusters are downstream targets of ESC-specific transcription factors, including OCT4, SOX2, and NANOG, which are essential for the self-renewal and maintenance of pluripotency in ESCs [47,63,66,67,68]. In both mouse and human ESCs, OCT4 and SOX2 bind to conserved promoter regions of miR-302 and activate its transcription. Furthermore, miR-302 directly suppresses differentiation marker genes, resulting in the high-level expression of pluripotent-related transcription factors, such as OCT4 [69]. miR-195 and miR-372a are also highly expressed in hESCs and maintain the proliferative capacity and self-renewal of hESCs through negative cell cycle modulators [70].

### 4.2. miRNA Clusters Are Involved in Stem Cell Differentiation

miRNAs are also involved in the initiation of stem cell differentiation by suppressing pluripotency-associated pathways. In human ESCs, post-transcriptional repression of *OCT4*, *SOX2*, and *KLF4* mediated by miR-145 leads to the suppression of self-renewal, and consequently the induction of differentiation [71]. Likewise, miR-134, miR-296, and miR-470 promote the differentiation of mouse ESCs by suppressing *Nanog*, *Oct4*, and *Sox2* [72,73]. miR-1305 induces the differentiation of human ESCs through the repression of *POLR3G*, which is an activator of the OCT4/NANOG pathway [74,75]. Other differentiation associated-miRNAs in mice are miR-34a, miR-100, and miR-137, which activate differentiation marker genes via the regulation of epigenetic mediators through targeting *Sirt1*, *Smarca5*, and *Jarid1b* mRNAs, respectively [76]. Furthermore, miR-27a and miR-24 silence self-renewal and promote differentiation by downregulating pluripotent-specific transcription factors and signal transducers of mESC self-renewal networks. The downregulation of Oct4 and Foxo1 and further suppression of signaling pathways through direct targeting of the LIF receptor *gp130* and the two important signal transducers *Smad3* and *Smad4* result in decreased expression of c-Myc, hence creating a mutual negative feedback loop to upregulate the expression of the miRNAs to maintain the differentiated state [77]. The let-7 miRNA family members also regulate the transition from self-renewing stem cells to differentiated cell types in both human and mouse ESCs. Contrary to the abovementioned pluripotent-specific miRNAs, let-7 miRNAs suppress the self-renewal of ESCs through cell cycle progression [78,79,80].

It is important to note that miRNAs might have context-dependent functions and can vary across different organisms and cell types. Numerous miRNAs contribute to the maintenance of early embryonic development and embryonic stem cell properties. In the following section, we highlight the well-studied stem-cell-specific miRNA clusters and compare their expression profiles and functions in humans, mice, rabbits, and chickens.

## 5. Major miRNA Clusters of ESCs

### 5.1. miR-302/367 Cluster

#### 5.1.1. Characteristics of Cluster

The polycistronic human miR-302/367 cluster is located in the intron of the LARP7 gene at the 4q25 region and consists of ten miRNAs: miR-302a-3p, miR-302a-5p, miR-302b-3p, miR-302b-5p, miR-302c-3p, miR-302c-5p, miR-302d-3p, miR-302d-5p, miR-367-3p, and miR-367-5p [64,65,81]. All 3′ mature miRNAs share the same seed sequence of 5′-AAGTGC-3′, except for miR-367 which harbors a slightly different seed sequence; however, they all have common mRNA targets [68,82]. The cluster is well conserved among the vertebrates (Figure 1). As it is shown in Figure 1, several species exhibit a common cluster structure, comprising four miR-302s (a–b) and one miR-367. Interestingly, the Aves class expresses an extra miRNA within the miR-302/367 cluster, such as gga-miR-1811 in chickens, which shares a highly similar seed sequence with miR-367, suggesting that it might be generated by the tandem duplication of miR-367 [81].

The miR-302/367 cluster’s promoter itself is a highly conserved region among vertebrates [47,83]. The cluster’s expression is regulated by ESC-specific transcription factors (OCT3/4, SOX2, KLF4, MYC, and NANOG) in mice and humans [68,84].

#### 5.1.2. miR-302/367 Expression in Mice, Humans, and Rabbits

In addition to cell cycle regulation, miR-302 cluster members modulate the self-renewal-related genes by repressing epithelial–mesenchymal transition and apoptotic pathways [85]. The miR-302/367 cluster is predominantly expressed in human ESCs, mouse epiblast stem cells (EpiSCs), and rabbit ES-like cells, whereas its expression in mouse ESCs is slightly lower [47,65,86]. In addition, it is highly expressed in human iPSCs and downregulated upon differentiation [87]. Overexpression of the miR-302/367 cluster in both mouse and human somatic cells results in cellular reprogramming the iPSC state and preserves the stemness properties of ESCs [88,89].

In rabbits, miR-302 expression begins at 3.5 dpc of pre-implanted embryos, whereas mouse embryos express miR-302 starting from day 6.5 dpc. The miR-302 cluster reaches the highest expression at the 6 dpc stage in rabbit embryos, and at 7.5 dpc in mouse embryos; its expression is rapidly downregulated by 8.5 dpc in mice [47,68].

#### 5.1.3. Gga-miR-302b-3p/5p Expression in Chicken Embryos and PGCs

Most chicken stem-cell-specific miRNAs are conserved with other vertebrate model species [90]. Gga-miR-302a is highly expressed in the chicken blastoderm and regulates the undifferentiated state of blastodermal cells and primordial germ cells (PGCs) by silencing *Sox11*, a somatic transcription factor [91]. It has also been reported that gga-miR-302b regulates glucose phosphate isomerase (GPI) expression; therefore, it modulates chicken PGC proliferation [92]. A recent study based on an inhibition assay demonstrated that gga-miR-302b (both -3p and -5p miRNAs) promotes the proliferation of PGCs and decreases apoptosis. In addition, the dual inhibition of -3p and -5p miR-302b caused an excessive increase in the number of apoptotic cells [93].

#### 5.1.4. Targets of miR-302/367 Cluster

To achieve the functional characterization of miRNAs, it is crucial to identify their target genes. Cell cycle regulators were the first-identified miR-302/367 targets both in human and mouse ESCs. In human ESCs, miR-302 modulates G1/S transition by directly targeting components of the cell cycle pathway, such as *CYCLIN D1* and *D2*, cyclin-dependent kinase 2 (*CDK2*), *CDK4*, *RB*, *E2F1*, *P130*, and *CDK6* [68]. In addition to repressing CYCLIN D1/D2 and other cell cycle components, it promotes S phase entry through an alternative pathway [94]. Likewise, in mouse ESCs, miR-302/367 suppresses the cyclin-dependent kinase inhibitor 1, also known as *p21*, thus inducing the G1 to S phase transition and cell proliferation [61]. It is also suggested that miR-302a modulates cell cycle progression in mouse ESCs through the negative regulation of *Lats2*, a tumor suppressor gene [95]. Sodium butyrate-mediated upregulation of the miR-302/367 cluster preserves the expression of key cell cycle regulators and supports self-renewal of human ESCs [96]. On the other hand, it also downregulates BNIP3L/Nix, thus inhibiting spontaneous apoptosis in hESCs [97]. *LEFTY1* and *LEFTY2* (Nodal inhibitors) are other prominent targets of miR-302 members. Post-transcriptional downregulation of *LEFTY* by miR-302 inhibits the expression of TGFβ/Activin/ NODAL family proteins and retains the pluripotency of hESCs through activating SMAD2/3 and inducing NANOG expression [98,99,100]. Similarly, *Lefty* has been identified as a direct target of miR-302 in rabbit ES-like cells by the transient inhibition of ocu-miR-302a [47]. *SMAD7* is a direct target of miR-367 in human pancreatic cancer cells and it promotes the invasion and metastasis of pancreatic cancer cells through the TGF-β signaling pathway [101]. miR-302/367 also promotes BMP signaling through direct targeting of BMP inhibitors, including *TOB2*, *DAZAP2*, and *SLAIN1*, thus maintaining pluripotency by repressing neural differentiation [102,103].

In addition to cell cycle and signaling pathway regulation, the miR-302 cluster modulates multiple key epigenetic regulators in somatic cells, including the lysine-specific histone demethylase enzymes *AOF1* and *AOF2* (also called *KDM1B* and *KDM1A*, respectively) and the methyl-CpG binding proteins *MECP1* and *MECP2*. Post-transcriptional suppression of these epigenetic regulators by miR-302 members induces global DNA demethylation and promotes reprogramming and iPS cell establishment [104]. Furthermore, *MBD2* (methyl-DNA binding domain protein 2) is a direct target of miR-302 to accomplish complete reprogramming of the iPS cell [105]. The main targets of miR-302/367 clusters are schematically presented in Figure 2.

### 5.2. C19miRNA Cluster

#### 5.2.1. Characteristics of C19M Cluster

C19MC (chromosome 19 miRNA cluster), one of the longest miRNA gene clusters in the human genome, is located on chromosome 19 and extends for about 100 kb. The cluster contains 46 highly homologous pre-miRNA genes, producing 56 mature miRNAs. Despite the high similarity of miRNAs in the cluster, they harbor 16 distinct seed sequences [106]. The cluster itself exhibits a unique genomic structure: most miRNA genes are flanked by ~400–700 bp Alu repeated sequences and short exons with highly repetitive DNA elements [107,108]. The Alu element reportedly mediates gene duplication events in this region, resulting in the expansive miRNA cluster and allowing for its high expression from the multiple copies [107,109]. Reciprocally, free Alu transcripts are post-transcriptionally suppressed by expressed miRNAs from the cluster; thus, the genome self-destruction caused by high rates of duplicative Alu transposition can be prevented [107]. In addition, hsa-miR-371/373 is located in close proximity to C19MC at about 20 kbp downstream of the cluster.

It has been suggested that C19MC miRNAs are primate-specific and have no orthologous regions in the mouse genome; however, we have previously identified three homologs in rabbits, including ocu-miR-512, ocu-miR-498, and ocu-miR-520e [47]. All three mature rabbit miRNAs presented high sequence similarity to their human homologs. They are located 5′ to the rabbit miR-290/295 cluster on the reverse strand of the pseudo-chromosome chrUn0226 in the same order as in humans (Figure 3a,b). Ocu-miR-520e shares the common consensus seed sequence of “AAGTGCT” with its human homologs, but ocu-miR-512 harbors a slightly different seed sequence.

#### 5.2.2. Expression of Human C19MC Cluster and Its Rabbit Homologs

The C19MC cluster miRNAs are trophoblast-specific and are highly expressed in the placenta [108]. In humans, C19MC is exclusively expressed from the paternal chromosome driven by an upstream promoter region. Mono-allelic expression of the C19MC cluster was demonstrated to be regulated through the DNA demethylation of the CpG-rich promoter region. Transcription of the C19MC cluster results in a primary transcript harboring the entire C19MC gene cluster, which is further processed into precursor miRNA and subsequently generates mature miRNAs via the Dgcr8-Drosha complex [110]. A recent study demonstrated that the pri-miRNA maturation of C19CM is tissue-specific and enhancer-mediated. A strong association of Drosha with the promoter/enhancer regions in human results is necessary for the efficient pri-miRNA maturation and to achieve a high expression level in placenta cells compared to the stem cells [111].

C19MC miRNAs are present in primary human-trophoblast-derived vesicles, particularly in the villous trophoblasts (VTs) [49,112]. They have also been detected in human extravillous trophoblasts (EVTs), where they may regulate migration [49]. Among C19MC-related miRNAs, ocu-miR-512 showed a high expression level in trophoblasts and hypoblasts of rabbit pre-implanted embryos [47].

In addition to the placenta, C19MC miRNAs are expressed in human ESCs and contribute to stemness [113,114]. Almost all C19MC miRNAs are active in naïve human ESCs, display high expression levels, and are epigenetically silenced in primed hESCs [115]. Unlike hESCs, rabbit ES-like cells do not exhibit a significant expression level of C19MC, which may reflect the primed state of rabbit ES-like cells [47]. The cluster is also expressed in human iPSCs and enhances reprogramming by suppressing epithelial-to-mesenchymal transition [116,117].

Furthermore, C19MC miRNAs are selectively expressed in various cancer types, such as breast cancer, hepatocellular carcinoma, embryonal brain tumors, infantile hemangioma, testicular germ cell tumors, parathyroid tumors, and thyroid adenomas. The overexpression of cancer-derived, circulating C19MC miRNAs, which commonly correspond to tumor size and proliferative state, makes them attractive potential biomarkers for diagnosis and treatment response monitoring [118,119,120,121,122,123,124,125].

#### 5.2.3. Targets of C19MC Cluster

Computational miRNA target prediction tools identified 4734 target genes for the C19MC cluster [114,115]. Among them, genes involved in chromatin structure modifications [114], the p53 pathway (including *CCNG2*, *CDKN1A*, *PMAIP1*, *TP53INP1*, and *ZMAT3*), and the extracellular matrix (ECM) [115] were enriched in hESCs.

miR-524 enhances somatic cell reprogramming by targeting *TP53INP1* (Tumor Protein P53 Inducible Nuclear Protein 1), promotes cell proliferation, and inhibits apoptosis [116]. Moreover, miR-524 downregulates *ZEB2* and *SMAD4* (epithelial–mesenchymal-transition-related genes) and promotes mesenchymal-to-epithelial transition, which is required for the initiation of reprogramming [126,127].

### 5.3. Mmu-miR-290/295, hsa-miR-371/373, and ocu-miR-290/295 Clusters

#### 5.3.1. Cluster Characteristics

The mmu-miR-290/295, hsa-miR-371/373, and ocu-miR-290/295 clusters are highly conserved in humans, mice, rabbits, chimpanzees, rats, dogs, and cows [128,129]; however, some differences are present in their structure, number of members, and genomic location.

The mmu-miR-290/295 cluster, which is the most abundant miRNA cluster in mouse ESCs [130], is coded by a 2.2 kb region located on chromosome 7 (Figure 4a). The single spliced primary transcript is processed into 14 mature miRNAs: miR-290-5p, miR-290-3p, miR-291a-5p, miR-291a-3p, miR-291b-5p, miR-291b-3p, miR-292-5p, miR-292-3p, miR-293-5p, miR-293-3p, miR-294-5p, miR-294-3p, miR-29-5p, and miR-295-3p [64,128]. All pre-miRNAs of the cluster share the same “AAGTGC” seed sequence, except for the miR-293 pre-miRNA (Figure 4b). The cluster’s transcription is regulated by a 332 nt intragenic enhancer region within the cluster [131]. Additionally, pluripotency-associated transcription factors, such as Oct4, Sox2, Snai, c-Myc, and Nanog, directly control the cluster expression by binding an upstream promoter element [66,132].

miR-371/373, a human ortholog of the mmu-miR-290/295 cluster, is located in close proximity to C19MC on chromosome 19 (Figure 4a) [64,125]. The hsa-miR-371/373 cluster is transcribed into four pre-miRNAs, pre-miR-371a, pre-miR-371b, pre-miR-372, and pre-miR-373, and further processed into six mature mRNAs: miR-371a-3p, miR-371a-5p, miR-371b-3p, miR-371b-5p, miR-372, and miR-373 [65,133,134]. The ESC-specific seed sequence “AAGTGC” can be found in all of the cluster’s miRNAs (Figure 4b).

Rabbit homologs of the mmu-miR-290/295 cluster are located on the reverse strand of a short pseudo-chromosome (chrUn0226) and are comprised of three mature miRNAs: ocu-miR290-5p, ocu-miR-292-3p, and ocu-miR-294-3p (Figure 4a). The ocu-miR-294-3p miRNA has also been mapped on chromosome 2 of the rabbit genome. Like hsa-miR-371/373, the ocu-miR290/295 cluster is located 3′ to the rabbit C19MC (Figure 4a). Interestingly, based on putative secondary structure prediction, ocu-miR-290-5p and ocu-miR-292-3p form a single pre-miRNA. Only ocu-miR-294 contains the stem-cell-specific seed sequence “AAGTGCT” (Figure 4b) [47].

#### 5.3.2. Evolution of mmu-miR-290/295, hsa-miR-371/373, and ocu-miR-290/295 Clusters

It is speculated that clustered miRNAs evolved via gene duplication, genomic rearrangements, and specific gene repressions. Evolving novel clusters might have comprised highly homologous miRNAs or could be joined with other miRNA families. In addition, miRNA family members can either be located in a cluster family or distributed randomly throughout the genome [135,136,137].

A cluster sequence and repeat analysis showed a close evolutionary relationship between human, mouse, and rabbit miRNA clusters. The miR-290/295 cluster has likely evolved from miR-290 or miR-291a and appears to be the evolutionary precursor. The individual pre-miRNA hairpin sequences of mmu-miR-290/295 and hsa-miR-371/373 clusters are homologous to each other. It appears that mouse miR-290/295 is the possible origin of the human miR-371/373 cluster, while the hsa-miR-371/373 cluster itself gave rise to another miRNA cluster, the hsa-miR-512 cluster, located in close proximity to its ancestor on human chromosome 19 [67]. Moreover, the enthalpy levels of pre-miR-372 and pre-miR-373 are similar to mmu-miR-291a, mmu-miR-291b, and mmu-miR-294 [67]. It has also been demonstrated that hsa-miR-373 is a member of the miR-520/373 family, composed of the hsa-miR-302/367, hsa-miR-371/373, and hsa-miR-520 clusters that share identical seed sequences [138,139,140].

Furthermore, according to the seed sequence and target interaction analysis, it seems that mmu-miR-290/295 and hsa-miR-371/373 share the same seed repertoire, given that in mice the cluster transcribes seven pre-miRNAs and in humans the cluster transcribes just three pre-miRNAs but the pre-miR-371 carries all the seeds to form the orthologs pre-miR-290, pre-miR-292, and pre-miR-293 in mice [141].

As mentioned above, the rabbit miR-290/295 cluster resides in only three miRNA members, representing a low evolutionary conservation with its homolog cluster in mice. Moreover, ocu-miR-290-5p and ocu-miR-292-3p form a single pre-miRNA, in contrast to the mouse homolog cluster which encodes three pre-miRNAs (pre-miR-290, pre-miR-291a, and pre-miR-292) in the same genomic region [47]. In addition, in proximity to the ocu-miR-290/295 cluster on the contig chromosome, there are three miRNAs of the C19MC cluster of ocu-miR-512, ocu-miR-520e, and ocu-miR-498, similar to the human and mouse cluster distribution. Hence, it is possible that these rabbit miRNAs located on the contig chromosome might recapitulate the evolution of human homologs and derive from the same ancestor by a duplication event [47].

#### 5.3.3. The Expression of mmu-miR-290/295, hsa-miR-371/373, and ocu-miR-290/295 Clusters

Mmu-miR-290/295 miRNAs are the first de novo expressed miRNAs in mouse embryos, starting at the two- to four-cell stage and exhibiting high expression in blastocysts [43,45,46]. The cluster is abundantly expressed in undifferentiated mESCs and maintains the pluripotent state of mESCs [66,130]. The overexpression of mmu-miR290/295 miRNAs under serum starvation retains the stem cell properties of mESCs and regulates the cell cycle by preventing a G1 stop and extending the survival of the stem cell population [67]. In addition, the simultaneous introduction of mmu-miR-290/295 along with Oct4, Sox2, and Klf4 (OSK) enhances the reprogramming of mouse embryonic fibroblasts into mouse iPSCs [142].

In contrast to mmu-miR-290/295, hsa-miR-371/373 expression in human ESCs is considerably lower and comprises only 1% of the total miRNA pool in hESCs [65,143]. Hsa-miR-372 and hsa-miR-373 display a high level of expression in human testicular germ cell tumors, indicating their oncogenic potential [133]. The introduction of hsa-miR-372 along with hsa-miR-302b enhanced the reprogramming efficiency of human fibroblasts [144].

The ocu-miR-290/295 miRNAs show low expression levels in rabbit ES-like cells, whereas they are highly expressed during preimplantation embryonic development, implying their potential role during early embryonic development [47].

Taken all together, the expression patterns of mmu- and ocu-miR-290/295 and hsa-miR-371/373 in ESCs and in early embryos suggest a prominent function in early embryonic development.

#### 5.3.4. miR-290/295 Cluster Targets

The mmu-miR-290/295 cluster is a well-described cell cycle regulator miRNA cluster. A short G1 phase and lack of a G1/S checkpoint are characteristics of the ESC cell cycle. The mouse miR-290/295 cluster regulates G1/S transition through the downregulation of cell cycle inhibitors such as *p21*, *Lats2*, *Wee1*, and *Fbxl5*, thus maintaining the pluripotent state of mouse ESCs [56,61,67,145]. Additionally, rapid G1/S transition promoted by mmu-miR-294 miRNA is Retinoblastoma gene (Rb)-independent. The accumulation of cells in G1 is regulated by direct targeting of the Rb family under nutrient starvation [94]. On the other hand, the mmu-miR-290/295 cluster reportedly represses S/G2 transition through *p53* and *Cyclin D1* [67,146]. Therefore, the mmu-miR-290/295 cluster contributes to pluripotency by accelerating G1 and extending the S phase in mESC.

Furthermore, mmu-miR-290/295 modulates DNA methylation in the pre-implantation embryos by the targeting of retinoblastoma-like 2 (*Rbl2*) and the subsequent inhibition of *Dnmt3b* [62,147]. On the one hand, pluripotent-associated transcription factors such as *Oct4*, *Sox2*, and *Nanog* either directly or indirectly regulate the transcription of DNMT3B. On the other hand, the core promoter region of the mmu-miR-290/295 cluster is a direct target for pluripotent-associated transcription factors. Altogether, pluripotent-associated transcription factors, the mmu-miR-290/295 cluster, *Rbl2*, and *Dnmt3b* build a regulatory network that regulates DNA methylation in early embryonic development and in ESCs [148]. Moreover, mmu-miR-290/295 directly targets *Dkk-1*, which is a Wnt pathway inhibitor, resulting in the upregulation of c-Myc, a downstream target of the Wnt signaling pathway [149,150]. Thus, mmu-miR-290/295 miRNAs partially favor pluripotency upon differentiation.

The mmu-miR-290/295 cluster together with miR-302/376 promote transition from naïve to primed pluripotency by enhancing the activity of the MEK pathway through the direct repression of *Akt1* [151]. In addition, mmu-miR-290/295 miRNAs are required for the incorporation of two core components of polycomb repressive complexes 2 (*Prc2*), *Ezh2*, and *Suz12* into promoters of bivalent differentiation genes in order to maintain the pluripotent state of mouse ESCs [152,153].

The mmu-miR-290/295 cluster also enhances the reprogramming of somatic cells and the quality of mouse iPSCs [89,142]. Mmu-miR-291 post-transcriptionally suppresses the methyltransferase *Ash1l* expression and consequently downregulates *HOX* genes, which in turn promote reprogramming through polycomb-mediated gene silencing [153].

In summary, the mmu-miR-290/295 cluster modulates both stemness and differentiation characteristics of mouse ESCs and embryos by regulating the cell cycle, de novo DNA methylation, apoptosis, and the transcription of pluripotent-associated transcription factors (Figure 5). To date, no target gene has been predicted for the rabbit miR-290/295 cluster. Certainly, target recognition and the investigation of functional characteristics are indispensable to discovering and describing the impact of the ocu-miR-290/295 cluster in early embryonic development and stem cell biology.

#### 5.3.5. Targets of hsa-miR-371/373 Cluster

To date, very little known is about how the hsa-miR-371-373 cluster can integrate into the pluripotency regulation of stem cells. The miR-372 miRNA together with miR-195 can partially rescue DICER knockdown phenotypes in hESCS but this is not sufficient to influence the cell cycle kinetics in wild-type hESCs. However, miR-372 represses *P21* (a CDK inhibitor); thus, it indirectly controls the G1/S transition and consequently regulates hESC division [70].

Due to its high level of expression and its potential biomarker capacity in different human tumor cells, its mechanism of action has been further dissected in relation to its oncogenic function. The miR-371/373 cluster activates WNT/β-catenin signaling through the repression of *DKK1*, which in turn induces cell growth and the invasive activity of tumor cells in humans [154]. However, it is not determined if the miR-371-373 cluster preserves the pluripotency of hESCs by activation of the WNT/β-catenin signaling pathway. In testicular germ cell tumor cells, the suppression of *LATS2* by miR-371/373 miRNAs prevents p53-mediated CDK inhibition and promotes tumorigenic growth [133]. Moreover, miR-373 miRNA represses *CD44*, thus leading to tumor migration and invasion in human breast cancer [138].

It also has been reported that miR-371/373 regulates glycolysis through the repression of *MBD2I*, resulting in promoting the expression of MYC and thus increasing human fibroblast reprogramming [155].

## 6. Conclusions

miRNAs are well-recognized, major post-transcriptional regulators of most cellular events including proliferation, cell cycle progression, differentiation, survival, and apoptosis in many cell types. Extensive research on the regulatory function of miRNAs indicates that miRNAs play crucial roles in stem cell properties. Pluripotent stem cells have enormous potential in the field of regenerative medicine, disease modeling, and new drug discoveries. In this review, we aimed to compare and discuss the regulatory functions of stem-cell-specific miRNA clusters and their action on the maintenance and regulation of ESC biology in vertebrates. We also discussed the expression profile of stem-cell-specific miRNAs in early embryogenesis, which may play an important role in the establishment and maintenance of pluripotent progenitor cells. Furthermore, we compared the stem-cell-specific human miRNAs with our other vertebrate model species.

Several miRNA clusters have been identified to regulate the stemness and pluripotency of stem cells. The most important miRNA clusters which are highly expressed in ESCs of vertebrates are miR-302/367 and miR-290/295 and its human homologue miR-371/373. They regulate the self-renewal and pluripotency of ESCs through the modulation of cell cycle progression. These miRNA clusters share a highly conserved seed sequence, “AAGUGCU” (referred to as stem-cell-specific miRNA) (Figure 2 and Figure 4), and induce the pluripotency of ESCs by direct targeting of the 3′ UTR of key ESC transcription factors, such as OCT4, SOX2, and NANOG. They also play a predominant role in vertebrate early development. Likewise, C19MC miRNAs are highly expressed in human ESCs, contribute to the stemness state of hESCs, and prevent their differentiation. Some members of C19MC miRNAs and their rabbit homologs also share the same stem-cell-specific “AAGTGCT” seed sequence (Figure 3).

The profiles of ESC-specific miRNA clusters and their relative functions in vertebrates which have been discussed through this review are summarized in Table 1.

This review gives outstanding insights into the cell cycle regulation of stem cells by miRNAs. A deep understanding of regulatory molecular networks in which miRNAs are interacting will greatly enhance our knowledge of miRNAs’ contribution to stem cell biology and therefore open promising avenues toward stem cell therapy.

## Figures and Tables

**Figure 1 genes-14-01434-f001:**
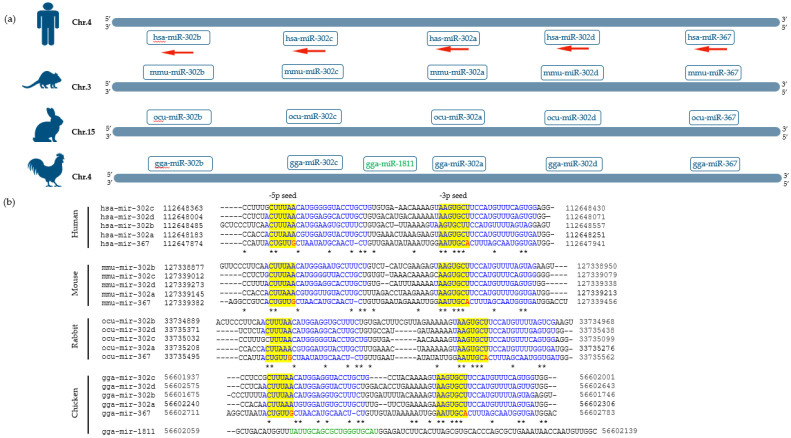
Genome organization and multiple sequence alignment of miR-302/367 precursor coding regions in humans, mice, rabbits, and chickens. (**a**) Humans, mice, rabbits, and chikens exhibit a common cluster structure, comprising four miR-302s and one miR-367. Interestingly, chikens express an extra miRNA, gga-miR-1811, within the miR-302/367 cluster. (**b**) Multiple sequence alignment of miR-302/367 precursor coding regions in humans, mice, rabbits, and chickens represent high similarity of this cluster. The seed sequences are highlighted in yellow; mismatches are shown in red. Mature -5p and -3p miRNA sequences are shown in blue. Mature sequence of gga-miR-1811 is shown in green. The asterisks indicate sequence homology.

**Figure 2 genes-14-01434-f002:**
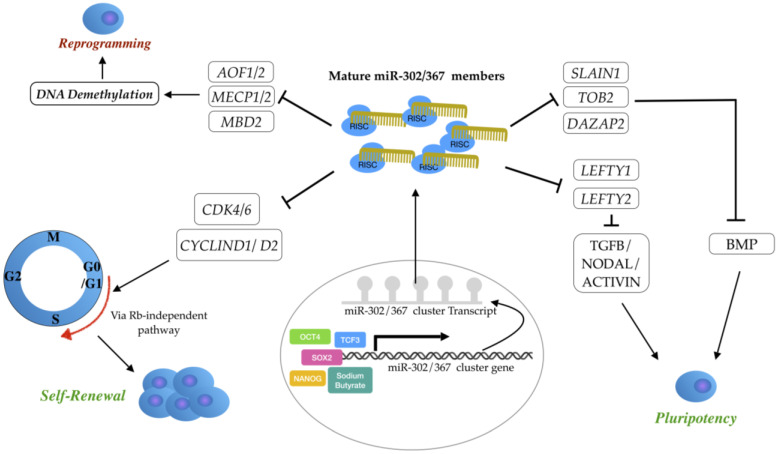
Main targets of miR-302/367 cluster. The miR-302-367 cluster plays a central role in maintaining ESC pluripotency and self-renewal by regulating different signaling pathways. The cytoplasmic mature miR-302/367 gene products positively regulate self-renewal by modulating G1/S transition through direct targeting of the cell cycle regulators, such as *CYCLIN D1/D2* and *CDK4/6*. Additionally, they promote S phase entry via an alternative Rb-independent pathway. These mature miRNA products also maintain the pluripotency of ESCs by targeting Nodal/Activin inhibitors, such as *LEFTY*. They also promote BMP signaling through direct targeting of BMP inhibitors (*TOB2*, *DAZAP2*, and *SLAIN1*), thus maintain pluripotency by repressing neural differentiation. Moreover, miR-302/367 members promote reprogramming and iPS cell establishment through induction of global DNA demethylation by post-transcriptional suppression of epigenetic regulators such as *AOF1*/*2*, *MECP1/2*, and *MBD2*.

**Figure 3 genes-14-01434-f003:**
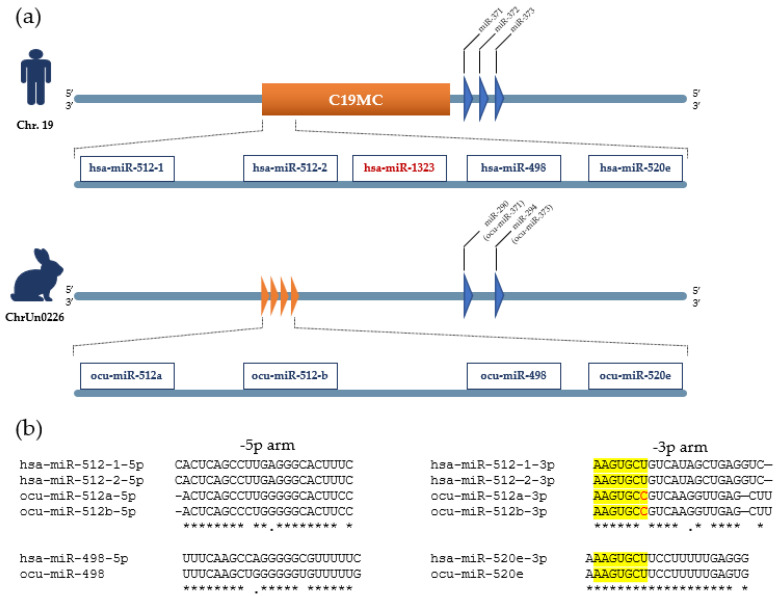
Genome organization and multiple sequence alignment of human C19MC cluster and three rabbit homologs. (**a**) Three rabbit homologues pre-miRNAs of human C19MC including ocu-miR-512, ocu-miR-498, and ocu-miR-520e are located on contig ChrUn0226 in same order as human C19MC on Chr.19. (**b**) Multiple sequence alignment represented high sequence similarity of rabbit mature miRNAs with their human homologues. Ocu-miR-520e shares the common consensus seed sequence of “AAGTGCT” with its human homologs, but ocu-miR-512 harbors a slightly different seed sequence. The seed sequences are highlighted in yellow. Mismatches in seed sequences are shown in red. The asterisks indicate sequence homology.

**Figure 4 genes-14-01434-f004:**
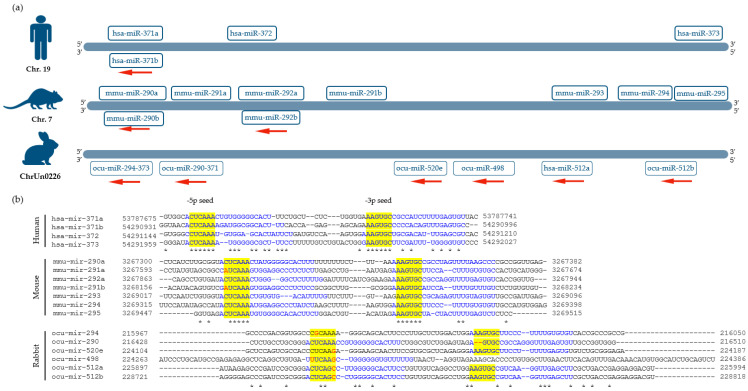
Genome organization and multiple sequence alignment of mmu-miR-290/295, ocu-miR-290/295, and hsa-miR-371/373 precursors in mice, humans, and rabbits. (**a**) miR-371/373, a human ortholog of the mmu-miR-290/295 cluster, is located in close proximity to C19MC on chromosome 19 and comprises of four pre-miRNAs. Rabbit homologs of the mmu-miR-290/295 cluster are located on the reverse strand of a short pseudo-chromosome (chrUn0226) and are comprised of three mature miRNAs. Similar to hsa-miR-371/373, the ocu-miR290/295 cluster is located 3′ to the rabbit C19MC. (**b**) All pre-miRNAs of mmu-miR-290/295 and hsa-miR-371/373 share ESC-specific seed sequence of “AAGTGC” except for mmu- miR-293 pre-miRNA. From rabbit cluster only ocu-miR-294 contains the same seed sequence. The seed sequences are highlighted in yellow. Mismatches in seed sequences are shown in red. Mature -5p and -3p miRNA sequences are shown in blue. The asterisks indicate sequence homology.

**Figure 5 genes-14-01434-f005:**
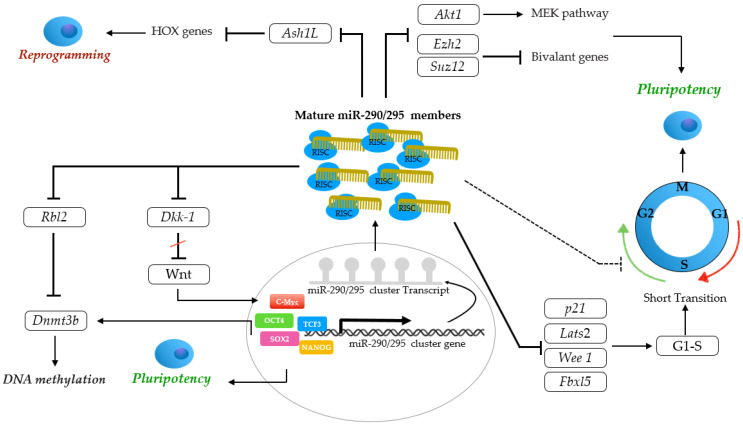
Main targets of miR-290/294 cluster. The miR-290/295 cluster modulates both stemness and differentiation characteristics of mouse ESCs and embryos by regulating cell cycle, de novo DNA methylation, apoptosis, and transcription of pluripotent-associated transcription factors. The cytoplasmic mature miR-290/295 members maintain pluripotency of mESC by accelerating G1 and extending S phase through direct targeting of cell cycle inhibitors such as *p21*, *Lats2*, *Wee1*, and *Fbxl5*, and also repressing S/G2 transition. These mature miRNAs also incorporate *Prc2*, *Ezh2* and *Suz12* into promoters of bivalent differentiation genes. The miR-290/295 cluster together with miR-302/376 promote transition from naïve to primed pluripotency by enhancing the activity of MEK pathway through direct repression of *Akt1*. These mature miRNAs partially favor pluripotency upon differentiation by direct targeting of *Dkk-1*, a Wnt pathway inhibitor, resulting in upregulation of c-Myc. The mature miR-290/295 gene products also modulate DNA methylation in the pre-implantation embryos by targeting of *Rbl2* and subsequent inhibition of *Dnmt3b*. The miR-290/295 cluster also enhances the reprogramming of somatic cells and the quality of mouse iPSCs through post-transcriptional suppression of *Ash1l* expression and consequently downregulation of *HOX* genes, which in turn promote reprogramming through polycomb-mediated gene silencing.

**Table 1 genes-14-01434-t001:** Summary of ESC-specific miRNAs and their relative function.

Cell Type	miRNAs	Expression Level	Function	Signaling Pathways	Targets
Mouse ESCs	miR-302/367 cluster	Low expression	Regulation of pluripotency, self-renewal, and reprogramming	Cdkn1a [61] TGF-β signaling pathway BMP signaling pathway	NA Lats2 [95]
	miR-290/295 cluster	High expression	Regulation of cell cycle progression and naïve pluripotency	ATM/ATR chek2-p53 pathway Wnt signaling pathway MEK Pathway activation	*Cyclin D1*, *p21*, *Lats2* [67,146] *Dkk-1* [150] *Akt1* [151]
			Early phases of differentiation	DNA methylation *HOX* gene inactivation	*Rbl2* [62,147] *Ash1l* [153]
Human ESCs	miR-302/367 cluster	High expression	Regulation of pluripotency, self-renewal, and reprogramming	Cyclin D1/D2 and Cdk2 [68] Nodal–Activin pathway	*LEFTY* [98,99,100] *BNIP3L/Nix* [97]
	miR-371/373 cluster	High expression	Cell cycle regulation Reprogramming	G1/S transition P53 pathway	*P21* [70] *MBD2* [155]
	C19MC	High expression in naïve hESCs and iPCs	Pluripotency maintenance	Chromatin structure modification pathway [114]	NA
			Prevent differentiation	ECM- related pathway [115]	NA
			Derivation and maintenance of human trophoblast stem cells	P53 pathway	*CCNG2 DKN1A PMAIP1 ZMAT3 TP53INP1* [115]
Rabbit ES-like cells	miR-302/367 cluster	High expression	Regulation of pluripotency and self-renewal	Nodal–Activin pathway	*Lefty* [47]
	miR-290/295 cluster	Low expression	Early embryonic development	NA	NA
	C19MC	Low expression	Early embryonic development	NA	NA
Chicken PGC line	miR-302b miR-302a	High expression	Cell proliferation Prevent differentiation	Glycolysis metabolism Somatic gene silencing	*Gpi3* [92] *Sox11* [91]

NA: not addressed.

## Data Availability

The data presented in this study are available on request from the corresponding author.

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
