# Peer review of "Pluripotency-Associated microRNAs in Early Vertebrate Embryos and Stem Cells"

_genes, 2023, doi:10.3390/genes14071434_

Round 1
Reviewer 1 Report (New Reviewer)
The review manuscript (Genes-2435055) by Maraghechi and colleagues presents an extensive overview of miRNAs and their roles in early embryo development and pluripotent stem cells. Overall, the paper is well written and logically organized, but contains a few grammatical errors and omissions that need to be addressed.
Firstly, the title needs to be revised. It is currently grammatically incorrect or not entirely accurate. Suggestions: Pluripotency-associated microRNAs in Vertebrate Preimplantation Embryos and Stem Cells; Pluripotency-associated microRNAs in Preimplantation Embryos and Vertebrate Stem Cells; Pluripotency-associated microRNAs in Preimplantation Embryos and Embryonic Stem Cells. Other?
In section 2. Overview of microRNA biogenesis, I believe an overview figure showing and describing the canonical and non-canonical pathways of miRNA biogenesis is warranted. I know, we see them in all the review papers but these are not simple processes, so a figure is warranted here.
Lines 64-66; “Many canonical miRNAs such as miR-302 and miR-290 clusters reportedly promote 64 embryonic stem cell (ESC) properties. MiR-320 and miR-702 are the two first discovered 65 non-canonical miRNAs expressed in mouse ESCs.” Citations need to be added to these statements. Please check through your entire manuscript to ensure that all statements of published material are cited.
In section 3. Role of the miRNAs in embryonic development, I think a few sentences indicating that miRNAs are known to be involved in embryo implantation and have been identified as potential markers of embryo quality and clinical outcomes is warranted here.
Lines 80-82; “…exhibited morphological abnormalities, such as suppressed OCT4 and brachyury expression, indicating deficient pluripotency and differentiation [30].” Suppressed OCT4 and Brachyury expression are not morphological abnormalities. They are abnormalities but not morphological abnormalities.
Lines 164-165; “They share the same seed sequence of 5′-AAGTGC-3′ UAAGUGCU expect for miR-367 which harbor slightly different seed sequence…”. “expect” should be “except”. Ensure that the seed sequence is correct. Shouldn’t the entire sequence be between 5’ and 3’? Suggestion: They share the same seed sequence of 5’-AAGTGCUAAGUGCU-3′ except for miR-367 which harbors a slightly different seed sequence.
Line 170; “…suggestion…” should be “…suggesting…”
Line 218; “…inducting…” should be “…inducing…”
Figures 2 and 5 should have a more extensive figure description. Concisely describe these figures please.
Figure 3. Add statement to description: Mismatches in seed sequences are shown in red.
Just minor revisions necessary (see above). Please review your manuscript to ensure quality.
Author Response
Answers to Reviewer 1 comments:
- Comment 1: Firstly, the title needs to be revised. It is currently grammatically incorrect or not entirely accurate. Suggestions: Pluripotency-associated microRNAs in Vertebrate Preimplantation Embryos and Stem Cells; Pluripotency-associated microRNAs in Preimplantation Embryos and Vertebrate Stem Cells; Pluripotency-associated microRNAs in Preimplantation Embryos and Embryonic Stem Cells. Other?
Answer to Comment 1:
Thank you for pointing this out. As suggested, we have corrected the title to “Pluripotency-associated microRNAs in Early Vertebrate Embryos and Stem Cells”.
- Comment 2: In section 2. Overview of microRNA biogenesis, I believe an overview figure showing and describing the canonical and non-canonical pathways of miRNA biogenesis is warranted. I know, we see them in all the review papers but these are not simple processes, so a figure is warranted here.
Answer to Comment 2:
Thank you for this suggestion. It would have been interesting to add this aspect. However, in the case of our review, it seems slightly out of scope as we mainly focused on miRNA profiling and their function. However, we have expanded the miRNA biogenesis, particularly the non-canonical pathway in section 2.
- Comment 3: Lines 64-66; “Many canonical miRNAs such as miR-302 and miR-290 clusters reportedly promote embryonic stem cell (ESC) properties. MiR-320 and miR-702 are the two first discovered 65 non-canonical miRNAs expressed in mouse ESCs.” Citations need to be added to these statements. Please check through your entire manuscript to ensure that all statements of published material are cited.
Answer to Comment 3:
We have modified the citation: Many canonical miRNAs such as miR-302 and miR-290 clusters reportedly promote ESC properties. MiR-320 and miR-702 are the two first discovered non-canonical miRNAs expressed in mouse ESCs [56]. Interestingly, miR-133a regulates the transcription of DNMT3B gene in human ESCs through a non-canonical pathway when it is translocated to the nucleus, while it regulates skeletal and cardiac muscle function via canonical target mRNA repression in cytoplasm [57].
We have also incorporated the paragraph to the section 4, lines 154-159.
- Comment 4: In section 3. Role of the miRNAs in embryonic development, I think a few sentences indicating that miRNAs are known to be involved in embryo implantation and have been identified as potential markers of embryo quality and clinical outcomes is warranted here.
Answer to Comment 4:
Thank you for this suggestion. It would have been interesting to explore this aspect. However, in the case of our review, the topic that we primarily focused on was the miRNA profile and the function of miRNAs during pre-implantation stages. However, the information you're requesting seems to be slightly beyond the scope of our discussion.
- Comment 5: Lines 80-82; “…exhibited morphological abnormalities, such as suppressed OCT4 and brachyury expression, indicating deficient pluripotency and differentiation [30].” Suppressed OCT4 and Brachyury expression are not morphological abnormalities. They are abnormalities but not morphological abnormalities.
Answer to Comment 5:
Thank you for pointing this out. We agree with this comment. Therefore, we have accordingly corrected the statement to clarify that the survived dicer 1 mutant embryos display morphological abnormalities and they do not express OCT4 and brachyury.
Lines: 111-113: Dicer 1 mutant mouse embryos exhibited a 50% lethality rate at 7.5 day embryonic stage;surviving embryos exhibited morphological abnormalities and suppressed Oct4 and brachyury expression, indicating an embryonic developmental arrest and deficient pluripotency [38].
- Comment 6: Lines 164-165; “They share the same seed sequence of 5′-AAGTGC-3′ UAAGUGCU expect for miR-367 which harbor slightly different seed sequence…”. “expect” should be “except”. Ensure that the seed sequence is correct. Shouldn’t the entire sequence be between 5’ and 3’? Suggestion: They share the same seed sequence of 5’-AAGTGCUAAGUGCU-3′ except for miR-367 which harbors a slightly different seed sequence.
Answer to Comment 6:
We have corrected the lines 222-223: All 3’ mature miRNAs share the same seed sequence of 5′-AAGTGC-3′ except for miR-367 which harbor slightly different seed sequence”.
- Comment 7: Line 170; “…suggestion…” should be “…suggesting…”
Answer to Comment 7:
We have accordingly modified the line 228-229: … which shares a highly similar seed sequence with miR-367 suggestion suggesting that … .
- Comment 8: Line 218; “…inducting…” should be “…inducing…”
Answer to comment 8:
We have corrected the line 278 as suggested: … SMAD2/3 and inducting inducing NANOG expression [97-99].
- Comment 9: Figures 2 and 5 should have a more extensive figure description. Concisely describe these figures please.
Answer to Comment 9:
Thank you very much for this suggestion. We agree with this comment. Therefore, we have added the following description to the figure legends of figure 2 and 5.
Figure 2: Main targets of miR-302/367 cluster. The miR-302-367 cluster play a central role in maintaining ESC pluripotency and self-renewal by regulating different signaling pathways. The cytoplasmic mature miR-302/367 gene products positively regulate self-renewal by modulating G1/S transition and promoting S phase entry through direct targeting of the cell cycle regulators, such as CYCLIN D1/D2, CDK4/6. These mature miRNA products also maintain the pluripotency of ESCs by targeting Nodal/Activin inhibitors such as LEFTY. They also promote BMP signaling through direct targeting of BMP inhibitors (TOB2, DAZAP2, and SLAIN1), thus maintain pluripotency by repressing neural differentiation. Moreover, miR-302/367 members promote reprogramming and iPS cell establishment through induction of global DNA demethylation by post-transcriptional suppression of epigenetic regulators such as AOF1/2, MECP1/2, and MBD2.
Figure 5. Main targets of miR-290/295 cluster. The miR-290/295 cluster modulates both stemness and differentiation characteristics of mouse ESCs and embryos by regulating cell cycle, de novo DNA methylation, apoptosis, and transcription of pluripotent-associated transcription factors. The cytoplasmic mature miR-290/295 members maintain pluripotency of mESC by accelerating G1 and extending S phase through direct targeting of cell cycle inhibitors like p21, Lats2, Wee1, and Fbxl5, and also repressing S/G2 transition. These mature miRNAs also in-corporate Prc2, Ezh2 and Suz12, into promoters of bivalent differentiation genes. The miR-290/295 cluster together with miR-302/376 promote transition from naïve to primed pluripotency by enhancing the activity of MEK pathway through direct re-pression of Akt1. These mature miRNAs partially favor pluripotency upon differentiation by direct targeting of Dkk-1, a Wnt pathway inhibitor, resulting in upregulation of c-Myc. The mature miR-290/295 gene products also modulate DNA methylation in the pre-implantation embryos by targeting of Rbl2 and subsequent inhibition of Dnmt3b. The miR-290/295 cluster also enhances the reprogramming of somatic cells and the quality of mouse iPSCs through post-transcriptional suppression of Ash1l expression and consequently downregulation of Hox genes, which in turn promote reprogramming through polycomb-mediated gene silencing.
- Comment 10: Figure 3. Add statement to description: Mismatches in seed sequences are shown in red.
Answer to Comment 10:
We accordingly added the following sentence to the figure legend of figure 3: “Mismatches in seed sequences are shown in red”.
We hope that this corrected version of our manuscript is appropriate for publication in the Special Issue “RNA Interference Pathways” of Genes.

Reviewer 2 Report (New Reviewer)
The review article by Maraghechi et al. summarized the functions of pluripotency-associated microRNAs in development and stem cells in four vertebrates. Overall, the review is comprehensive and informative. One major suggestion I have is to reconsider how to include the chicken results. This review is aimed to establish the conversation of those miRNA clusters in genomic structures and biological functions in vivo (development) and in vitro (cultured stem cells). However, in vivo functions of miRNAs in chicken were only validated in primordial germ cells (PGCs), while no parallel comparisons (miRNA functions in PGCs) were described in other vertebrates. For in vitro comparison, there is also no equivalent cell culture model in chicken as ES/ES-like cells in human/mouse/rabbit. Thus, the functional conservation is not that clear for all four vertebrates. If the authors want to keep the structure and results in this review, it will be better to include the PGC results in mammalian species and related the species and phenotypes more clearly. In that case, Line #31-33 can be “the impact of stem-cell specific miRNA clusters on the maintenance and regulation of early embryogenesis in vertebrates, particularly in human, mouse, rabbit and chicken, as well as their functions in pluripotent stem cells.” And adjust the rest of the manuscript.
Other issues:
1. The title. Most miRNA-deficient phenotypes were in post-implantation embryos. The strongest evidence is that the Dgcr8 knockout embryos can develop until the blastocyst stage (Blelloch lab, also see point 7).
2. Line 35: Pol III doesn’t produce pri-miRNA and should be removed. Although few miRNAs can be derived from Pol III-transcribed small RNAs, they should be regarded as non-canonical and can be addressed later.
3. Line 64-69 don’t seem to belong this section. Maybe incorporate them into the first paragraph of Section 4?
4. Expand the non-canonical miRNA paragraph (Line 59-63) a bit and give some examples.
5. Line 71-74: separate lin-4 and let-7; expand a bit of the history of discovery. lin-4 is the “first known” miRNAs, and let-7 was immediately recognized as a conserved miRNA cluster and its knockout exhibited strong phenotype, triggering high-level interest.
6. Unify “ES cells” and “ESCs”.
7. Section 3 is better to distinguish the expression and the phenotype, as well as the pre-implantation and post-implantation stage. As mentioned in point 1, the phenotype of Dgcr8 knockout argues that miRNAs are likely to exhibit their functions in post-implantation embryos.
8. Ref 56 is not correctly summarized in Line #138-140.
9. Line 164, should only present the 6-nt seed sequence.
10. Some texts in figures are fuzzy.
11. Line 498-504 are not correct for typesetting.
12. The nomenclature of genes/proteins should be unified (such as Ago and Argo) and species-specific. Also, since miRNAs targets mRNAs, the miRNA targets should be Italic to demonstrate the difference between proteins and RNAs. For example, both line 138 and 208 list miRNA targets, which should all be Italic. Here are some suggestions:
a) Human genes/proteins: all capital, regular (e.g. CDK4)
b) Mouse/rabbit genes/proteins: first capital, regular (e.g. Cdk4)
c) mRNAs/miRNA targets: Italic (e.g. CDK4 for human and Cdk4 for mouse)
Author Response
Answers to Reviewer 2 comments:
- Comment 1: The title. Most miRNA-deficient phenotypes were in post-implantation embryos. The strongest evidence is that the Dgcr8 knockout embryos can develop until the blastocyst stage (Blelloch lab, also see point 7).
Answer to Comment 1:
Thank you very much for pointing this out. Our focus in this manuscript were mostly on miRNA profiling in preimplantation stages in both section 3 and 5. However, we also discussed the miRNA function in post-implantation. Therefore, as suggested we agree to change the title to “Pluripotency-associated microRNAs in Early Vertebrate Embryos and Stem Cells”.
- Comment 2: Line 35: Pol III doesn’t produce pri-miRNA and should be removed. Although few miRNAs can be derived from Pol III-transcribed small RNAs, they should be regarded as non-canonical and can be addressed later.
Answer to Comment 2:
Thank you for this suggestion. However, the Line 35: “MiRNA genes are transcribed by RNA polymerase II/III into long primary transcripts (pri-miRNA)” refers to both canonical and non-canonical pathways which was separately described in the two followed paragraphs. However, we have accordingly specified the Pol types for each pathway:
Lines 49-50: In the canonical pathway, pri-miRNAs are mostly transcribed by RNA Polymerase II [5].
Lines 66-67: Non-canonical miRNAs are also derived from RNA Polymerase II; however, few miRNAs can be transcribed from RNA Polymerase III.
Line 70-71: MiRNAs derived from endo-shRNAs and tRNA precursors are transcribed by RNA Polymerase III [24,27].
- Comment 3: Line 64-69 don’t seem to belong this section. Maybe incorporate them into the first paragraph of Section 4?
Answer to Comment 3:
Thank you for this suggestion. We have accordingly incorporated them into section 4, lines 152-159: “To date, many studies have demonstrated the regulatory function of miRNAs (both canonical and non-canonical) in the self-renewal and differentiation of ESCs and iPSCs. Many canonical miRNAs such as miR-302 and miR-290 clusters reportedly promote embryonic stem cell (ESC) properties. MiR-320 and miR-702 are the two first discovered non-canonical miRNAs expressed in mouse ESCs [56]. Interestingly, miR-133a regulates the transcription of DNMT3B gene in human ESCs through a non-canonical pathway when it is translocated to the nucleus, while it regulates skeletal and cardiac muscle function via canonical target mRNA repression in cytoplasm [57].”
- Comment 4: Expand the non-canonical miRNA paragraph (Line 59-63) a bit and give some examples.
Answer to Comment 4:
We have accordingly expanded the non-canonical pathway.
Lines 66-71: Non-canonical miRNAs are also derived from RNA Polymerase II; however, few miRNAs can be transcribed from RNA Polymerase III. The non-canonical miRNA biogenesis occurs trough a Drosha/DGCR8 and/or Dicer-independent manner, when miRNAs originate from introns (mirtrons) [22,23] or other small RNAs, such as snoRNAs, shRNAs, tRNAs, tRNase Z, and endo-siRNAs [24-28]. MiRNAs derived from endo-shRNAs and tRNA precursors are transcribed by RNA Polymerase III [24,27].
Lines 72-88: In Drosha/DGCR8-independent pathway (also called mirtron pathway), introns are primarily processed in the nucleus by spliceosome and a debranching enzyme that produce short hairpins called mirtrons. These hairpins are then exported to the cyto-plasm by Exportin-5 and cleaved by Dicer [22,23]. Even though, the majority of intron in vertebrates are elongated from several kilobases up to megabases in length, there are short introns of 50–150 nt in length which may form hairpin structure [29,30]. Hence, mitrons constitute only a small fraction of mammalian miRNAs. There are other sub-classes of miRNAs with a potential pre-miRNA-like hairpin which are derived by al-ternative Drosha/DGCR8-independent pathway. The processing of snoRNAs, shRNAs, tRNAs, tRNase Z, and endo-siRNA precursors to miRNAs are also independent of Drosha/DGCR8 [24-28].
In the Dicer-independent pathway, produced pri-miRNA are processed by Drosha into pre-miRNAs. However, these pre-miRNAs are not suitable substrates for Dicer; therefore, they are loaded into Ago2 instead, and Ago2-dependent cleavage pro-duces an intermediate 3′ end. Final maturation occurs as the 3′-5′ trimming of the 5′ strand completes by a cellular nuclease [31,32]. In Dicer-independent, tRNase Z-dependent pathway precursor tRNA is cleaved by tRNase Z at the 3’-end [33].
- Comment 5: Line 71-74: separate lin-4 and let-7; expand a bit of the history of discovery. lin-4 is the “first known” miRNAs, and let-7 was immediately recognized as a conserved miRNA cluster and its knockout exhibited strong phenotype, triggering high-level interest.
Answer to Comment 5:
We have accordingly separate line-4 and line-7.
Lines 96-104: Many miRNAs function during early embryonic development. The first known miRNA, lin-4 play a crucial role in developmental timing of stage-specific cell lineages in C. elegans [34]. Shortly after discovery of line-4, let-7 was recognized as an essential miRNA to control pattern of temporal development of larval cell fates during the adult stage and which acts as a temporal switch between larval and adult fates. Loss of let-7 gene activity leads to abnormalities in the pattern of larval development by reiteration of larval cell fates during the adult stage, while increased let-7 gene dosage causes precocious expression of adult fates during larval stages [35].
- Comment 6: Unify “ES cells” and “ESCs”.
Answer to Comment 6:
Thank you for this suggestion. We have accordingly changed “ES cells” into “ESCs” in entire manuscript.
- Comment 7: Section 3 is better to distinguish the expression and the phenotype, as well as the preimplantation and post-implantation stage. As mentioned in point 1, the phenotype of Dgcr8 knockout argues that miRNAs are likely to exhibit their functions in post-implantation embryos.
Answer to Comment 7:
Thank you for this suggestion. In section 3, we mentioned that the impact of miRNAs in embryonic development has been demonstrated in different organisms either by dissecting miRNA pathway or by functional and expression pattern analysis of miRNAs.
In second paragraph of section 3 (lines 106-117), we discussed the dissection of miRNA pathway and we stated the pos-implantation stage for all mutations: Deletion of Dicer, Dgcr8 or Ago2, induce early post-implantation lethality (lines 109-110).
In third paragraph (lines 118-150) we focused on functional and expression pattern analysis of miRNAs from zygote to blastocyst, and mentioned preimplantation stage when the exact embryonic stage was not specified.
- Comment 8: Ref 56 is not correctly summarized in Line #138-140.
Answer to Comment 8:
Thank you for pointing this out. There has been a technical error which has changed the original text before submission. We have corrected the paragraph related to ref 56 (in revised version ref 69) which can be found in lines 183-187: In both mouse and human ESCs, OCT4 and SOX2 bind to a conserved promoter regions of miR-302 and activate its transcription. Furthermore, miR-302 directly suppresses differentiation marker genes resulting in high-level expression of pluripotent related transcription factors, like OCT4 [69].
- Comment 9: Line 164, should only present the 6-nt seed sequence.
Answer to Comment 9:
Thank you for pointing this out. We have corrected the lines 222-224: All 3’ mature miRNAs share the same seed sequence of 5′-AAGTGC-3′ except for miR-367 which harbor slightly different seed sequence.
- Comment 10: Some texts in figures are fuzzy.
Answer to Comment 10:
There is a size reduction in Word, which is not allowing us to insert higher resolution in the document. However, high-resolution images for final version have been uploaded separately.
- Comment 11: Line 498-504 are not correct for typesetting.
Answer to Comment 11:
Thank you for suggestion. We have modified the typesetting.
Lines 578-579: Author Contributions: Conceptualization, E.G., P.M., and MT.S.A.; writing - original draft preparation: P.M, MT.S.A.; and E.G.; writing review and editing: B.L., A.E., and R.T.; visualization: MT.S.A., N.T.S.; supervision: E.G.; funding acquisition: E.G. All authors have read and agreed to the published version of the manuscript.
- Comment 12: The nomenclature of genes/proteins should be unified (such as Ago and Argo) and species specific. Also, since miRNAs targets mRNAs, the miRNA targets should be Italic to demonstrate the difference between proteins and RNAs. For example, both line 138 and 208 list miRNA targets, which should all be Italic. Here are some suggestions:
- a) Human genes/proteins: all capital, regular (e.g. CDK4)
- b) Mouse/rabbit genes/proteins: first capital, regular (e.g. Cdk4)
- c) mRNAs/miRNA targets: Italic (e.g. CDK4 for human and Cdk4 for mouse)
Answer to Comment 12:
Thank you very much for this suggestion. We have unified them in entire manuscript including the text, table, and figures according to the suggested nomenclature of genes/proteins. However, whenever we talked about human together with mouse or/and rabbit, we have used the nomenclature of human.
We hope that this corrected version of our manuscript is appropriate for publication in the Special Issue “RNA Interference Pathways” of Genes.

Reviewer 3 Report (New Reviewer)
The review by Maraghechi et al. discusses the roles of a number of microRNAs during early development and cultured ESCs. The text is mostly focused on a reduced number of microRNA clusters with a comparative perspective between mammals and chicken. I initially thought that the topic was very interesting but I was really disturbed by the lack of order, focus and wrong concepts presented in the review. Briefly,
1. Why the text focuses on the 290, 302 and 371 clusters? There are many more, and probably more important or interesting microRNAs involved in preimplantation embryos and stem cells in vertebrates? The list of long but I could mention a few of them: miR-34, miR-203, let7, etc.
2. The organization of the review can be hugely improved. Many concepts and data are repeated multiple times making the text really long and repetitive. For example, the expression pattern is repeated in many different paragraph, the targets are also repeated etc. There are so many examples of repetition that I cannot list them here.
Lines 64-69. Not needed here and repeated many times thereafter.
3. Some concepts that are wrong or to revise
Line 36. The majority of miRNAs are intergenic, not intragenic.
Lines 39-40. Not all members of a cluster same the same seed sequence (see for instance the example of C19).
Lines 55-56. Not sure this is always true.
Lines 134-135 (and some other instances later in the text): these miRNA clusters “induce pluripotency by direct targeting the 3’UTR of key ESC transcription factors, such as OCT4, SOX2 and NANOG”.. You cannot induce pluripotency by targeting (and therefore degrading or destabilizing) the mRNA of key TFs.
Line 137-138: “miR-302 directly suppresses differentiation marker genes (including cdk2, cdk4, akt1…” None of these are differentiation marker genes (they are actually the opposite).
Lines 161-171 ( and several others through the text). The authors mention all the mature miRNAs in a cluster (including all the 5p and 3p sequences) and indicate that they all have the same seed sequence, which is impossible for the 5p versus 3p miRNAs.
Lines 205-208. “miR-302 promotes S-phase entry by direct targeting components of the cell cycle pathway, such as cyclin D1 and D2, CDK2, CDK4, etc.” This is again wrong. If you target these genes, you prevent S-phase entry.. (Figure 2 is also wrong)
In general, is this review goes through the publication process I would suggest:
- A complete reorganization presenting perhaps a) discovery; b) genomic organization; c) expression pattern and d) targets and function.
- Revise all the concepts. Some of the wrong concepts above invalidate the whole review.
- Review data with criticism, do not use computational predictions, elaborate whether the data is some papers make sense or not
- Focus to something more specific (either early (then define) in vivo, stem cells, or blastocyst stage, or trophoblast, etc.), why some miRNAs and not others etc
N/A
Author Response
Answers to Reviewer 3 comments:
- Comment 1a: Why the text focuses on the 290, 302 and 371 clusters?
Answer to Comment 1a:
In this review, we focused on impact of well-studied stem cell-specific miRNA clusters, miR-290, its human homolog miR-371, and miR-302 on the maintenance and regulation of early embryogenesis and embryonic stem cell pluripotency in vertebrates; particularly in human, mouse, rabbit, and chicken.
Early embryonic development can be viewed as a process in which self-renewal and differentiation properties of stem cells are altered. Many studies showed that miRNAs are involved in epigenetic reprogramming, maternal RNA clearance, transcriptional and translational landscape changes, as well as the signal transduction which are required for the proper development of early embryos, therefore they modulate biological properties of stem cells. The above-mentioned miRNA clusters are expressed in both early embryos and ESCs and involved in early embryogenesis and tuning ESCs. Understanding the functions of these stem-cell specific miRNAs give a more comprehensive picture of embryonic development and control stem cells self-renewal and pluripotency.
In addition, our laboratory research work focuses on rabbit, mouse and chicken stem cell specific miRNA investigation. In this review we wanted to highlight the similarity and difference between them and compare them with human miRNAs.
To point out the importance of the clusters which we are focusing on, we have also added a description on section 3.
Lines 211-215: It's important to note that miRNAs might have context-dependent function and can vary across different organisms and cell types. Numerous miRNAs contribute to the maintenance of early embryonic development and embryonic stem cell properties. In the following section we highlighted the well-studied stem cell-specific miRNA clusters and compared their expression profile and function in human, mouse, rabbit, and chicken.
- Comment 1b: There are many more, and probably more important or interesting microRNAs involved in preimplantation embryos and stem cells in vertebrates? The list of long but I could mention a few of them: miR-34, miR-203, let7, etc.?
Answer to Comment 1b:
Thank you for this suggestion. We accordingly added some other important microRNAs involved in preimplantation embryos and stem cells in vertebrates to the section 3 and 4. We have also incorporated the last paragraph of section 2 (lines 64-69) into section 4. The role of miR-34a has been explained in section 4.2 (Lines: 197-200: Other differentiation associated-miRNAs in mouse are miR-34a, miR-100, and miR-137 which activate differentiation marker genes by regulation of epigenetic mediators through targeting Sirt1, Smarca5, and Jarid1b mRNAs, respectively [76]). However, miR-203 is mostly involved in epithelial-to-mesenchymal transition and fast muscle differentiation during later embryonic stage or in differentiation of iPSCs. Since our focus is mostly on pre-implantation embryonic development and embryonic stem cells, it seems slightly out of scop.
Modifications:
Section 3:
Lines 96-104: The first known miRNAs, lin-4, and let-7 play a crucial role in developmental timing of stage-specific cell lineages the temporal control of larval development in C. elegans [34]. Shortly after discovery of line-4, let-7 was recognized as an essential miRNA to control pattern of temporal development of larval cell fates during the adult stage which acts as a temporal switch between larval and adult fates. Loss of let-7 gene activity leads to abnormalities in the pattern of larval development as a result of reiteration of larval cell fates during the adult stage, while overexpression of let-7 gene causes early presence of adult cell fate during larval stages [35].
Lines 145-150: MiR-21 is expressed in both human and mouse oocytes and blastocysts [53,54]. It is associated with oocyte maturation, blastocyst formation, and preimplantation embryonic development [54]. Let-7 family also play crucial role in mammalian embryonic development. Expression of Let-7 in mouse embryos regulates inner cell-mass cell fate de-termination and reinforces blastocysts formation [55].
Section 4:
Lines 152-159: To date, many studies have demonstrated the regulatory function of miRNAs (both canonical and non-canonical) in the self-renewal and differentiation of ESCs and iPSCs. Many canonical miRNAs such as miR-302 and miR-290 clusters reportedly promote ESC properties. MiR-320 and miR-702 are the two first discovered non-canonical miRNAs expressed in mouse ESCs [56]. Interestingly, miR-133a regulates the transcription of DNMT3B gene in human ESCs through a non-canonical pathway when it is translocated to the nucleus, while it regulates skeletal and cardiac muscle function via canonical target mRNA repression in cytoplasm [57].
Lines 187-189: MiR-195 and miR-372a are also highly expressed in hESCs and maintain the proliferative capacity and self-renewal of hESCs through negative cell cycle modulators [70]
Lines 206-210: The let-7 miRNA family members also regulate the transition from self-renewing stem cells to differentiated cell types in both human and mouse ESCs. Contrary to above mentioned pluripotent specific miRNAs, let-7 miRNAs suppress self-renewal of ESCs through cell cycle progression [78-80].
- Comment 2:
- a) The organization of the review can be hugely improved. Many concepts and data are repeated multiple times making the text really long and repetitive. For example, the expression pattern is repeated in many different paragraph, the targets are also repeated etc. There are so many examples of repetition that I cannot list them here.
Answer to Comment 2a:
Thank you for the suggestion. We have accordingly modified the repeated information through the manuscript.
Section 3:
Lines 121-126: Remarkably, certain stem cell-specific miRNAs, such as miR-290 cluster members, are the first de novo expressed miRNAs in early embryogenesis the 2- to 4-cell stage mouse embryos and are significantly up-regulated in the blastocyst [43,45,46]. Similarly, in rabbit, the expression of ocu-miR-290 cluster members begins at early embryonic stages 4-cell stage and shows high expression throughout the blastocyst stage [47].
Lines 126-130: Furthermore, ocu-miR-512-5p, which is evolutionarily related to miR-290 and its hu-man homolog miR-371 is also expressed in early stages of rabbit embryonic development was expressed in rabbit embryos between the 8-cell stage and 4.5 days post-coitum (dpc), and its expression substantial decreased at 6 dpc stage [47].
Lines 142-144: Contrarily, the expression of ocu-miR-302 cluster members were very low during early embryonic stages, and their expression was initiated at 3.5 dpc stage followed by up-regulation at the blastocyst stage. [47].
- b) Lines 64-69. Not needed here and repeated many times thereafter
Answer to Comment 2b:
We have accordingly removed the paragraph from mentioned part.
Comment 3: Some concepts that are wrong or to revise?
Answer to Comment 3:
- Line 36. The majority of miRNAs are intragenic and are transcribed from their own promoter.
Thank you for this suggestion. We have accordingly revised and corrected the phrase. We have also corrected the paragraph and added the related references.
Lines 36-43: A significant percentage of miRNAs are intragenic residing withing their host gene’s intronic or/and exonic regions [7]. They are transcribed from introns, and occasionally from exons of protein-coding genes; sharing the host gene’s ex-pression regulators. Whereas, intergenic miRNAs are located between genes and transcribed independently from their own promoters [6,8]. However, some intergenic miRNAs can also be co-transcribed with their neighboring genes [9].
- Lines 39-40. A ‘cluster’ is a group of miRNAs that share the same seed sequence and are transcribed from the same genomic loci.
We have revised the above sentence with corresponding references as following:
Lines 43-47: Many miRNAs are localized as a cluster on a same chromosome in a close proximity of maximum 10 kb [10,11]. They are co-transcribed from the same genomic loci as a single polycistronic transcript [12,13]. They alsoshare the same seed sequence, therefore sharing common target mRNAs and biological function [10,14].
- Lines 55-56. Not sure this is always true.
A perfect base pairing results in target degradation, while imperfect base pairing results in translational silencing of target mRNA.
Thank you for this suggestion. However, the statement is generally true. The extent of complementarity between the seed sequence of a miRNA and its target mRNA determines the outcome of miRNA-mediated regulation. The above statement has been stated in several research publication.
Selbach, et al., measured changes in production of ~5,000 proteins altogether using combined pSILAC with state-of-the-art mass-spectrometry-based proteomic. They have demonstrated that only seed-containing mRNAs with at least one mismatch were, overall, repressed at the protein level. In contrast, protein production from seed-containing mRNAs with perfect base pairing from nucleotides 9 to 11 and mRNAs lacking seeds was indistinguishable (DOI: 10.1038/nature07228).
Members of a class represented by Arabidopsis miRNA 39 interact with perfect complementarity and appear to mimic siRNA function to guide cleavage. Support for this concept also comes from the finding that engineered miRNA-target combinations with perfect complementarity result in target RNA cleavage (DOI: 10.1017/s13558382020240326 and 10.1126/science.1073827 and 10.1038/35005107).
It has been shown that, in human cell extracts, the miRNA let-7 naturally enters the RNAi pathway, which suggests that only the degree of complementarity between a miRNA and its RNA target determines its function. A miRNA will direct destruction of the target mRNA if it has perfect or near-perfect complementarity to the target. The endogenous let-7 in the HeLa cytoplasmic extract is associated with RISC and enters the RNAi pathway which leads to direct cleavage of the let-7 complementary RNA target (DOI: 10.1126/science.1073827).
- Lines 134-135 (and some other instances later in the text): these miRNA clusters “induce pluripotency by direct targeting the 3’UTR of key ESC transcription factors, such as OCT4, SOX2 and NANOG”. You cannot induce pluripotency by targeting (and therefore degrading or destabilizing) the mRNA of key TFs.
Thank you for pointing this out. We have accordingly corrected the above statement.
Lines 179-182: These clusters are downstream targets of ESC-specific transcription factors and induce pluripotency by direct targeting the 3′ UTR of key ESC transcription factors, such as including OCT4, SOX2, and NANOG which are essential for the self-renewal and maintenance of pluripotency in mESCs[47,63,66-68].
- Line 137-138: “miR-302 directly suppresses differentiation marker genes (including cdk2, cdk4, akt1...” None of these are differentiation marker genes (they are actually the opposite).
Thank you for pointing this out. There has been a technical error which has changed the original text before submission. We have corrected the paragraph which can be found in lines 184-187: Furthermore, miR-302 directly suppresses differentiation marker genes resulting in high-level expression of pluripotent related transcription factors, like OCT4 [69].
- Lines 161-171 ( and several others through the text). The authors mention all the mature miRNAs in a cluster (including all the 5p and 3p sequences) and indicate that they all have the same seed sequence, which is impossible for the 5p versus 3p miRNAs.
Thank you for the suggestion. We have accordingly revised and corrected the text.
Lines 222- 224: All 3’ mature miRNAs share the same seed sequence of 5′-AAGTGC-3′ except for miR-367 which harbor slightly different seed sequence;
Lines 325-327: They Ocu-miR-520e share the common consensus seed sequence of 5′-AAGTGC-3′ with its their human homologs, but ocu-miR-512 harbors slightly different seed sequence.
- Lines 205-208. “miR-302 promotes S-phase entry by direct targeting components of the cell cycle pathway, such as cyclin D1 and D2, CDK2, CDK4, etc.” This is again wrong. If you target these genes, you prevent S-phase entry
Thank you for the suggestion. However, the statement is generally true: miR-302 can promote S-phase entry by directly targeting components of cyclin D1 and D2, CDK2, and CDK4. It has been reported that miR-302 play a role in cell cycle regulation, specifically in promoting the transition from the G1 to S phase of the cell cycle. MiR-302 is known to directly target and suppress the expression of cyclin D1, cyclin D2, CDK2, and CDK4, which are critical regulators of cell cycle progression. By targeting these components, miR-302 inhibits their expression, leading to a disruption in the regulation of the G1/S transition. This, in turn, promotes the entry of cells into the S phase of the cell cycle (Doi: 10.1042/BJ20071635 and 10.1016/j.stem.2011.03.001).
However, we have accordingly modified the above sentences which can be found in lines 265-268: In human ESCs, miR-302 modulates G1/S transition and consequently promotes S phase entry by directly targeting components of the cell cycle pathway, such as CYCLIN D1, and D2, cyclin-dependent kinase 2 (CDK2), CDK4, RB, E2F1, P130 and CDK6 [68].
- (Figure 2 is also wrong)
Thank you for this suggestion. There were slightly little errors which we have modified and corrected Figure 2.
In general, is this review goes through the publication process I would suggest:
A complete reorganization presenting perhaps a) discovery; b) genomic organization; c) expression pattern and d) targets and function.
Our review follows almost a similar structure. Only in section 5 we discussed the two main miRNA clusters separately and we compared them in human, mouse, rabbit, and chicken to give a comprehensive and comparative perspective of each cluster’s location, expression, function, and targets in mentioned organisms. However, we also followed the same organization structure in section 5 and repeated for each cluster separately.
Revise all the concepts. Some of the wrong concepts above invalidate the whole review.
We have revised and improved our article according to the reviewers' comments. We believe that there are no parts left in the report that contain incorrect statements.
Review data with criticism, do not use computational predictions, elaborate whether the data is some papers make sense or not
In our article, we have only described specific microRNA target sites, referring to the literature references, where target sites have been experimentally validated.
We hope that this corrected version of our manuscript is appropriate for publication in the Special Issue “RNA Interference Pathways” of Genes.

Round 2
Reviewer 3 Report (New Reviewer)
I appreciate the effort of the authors and the manuscript is much improved. However, the following text, figure 2 and answer (see below) is wrong. One of the DOIs cited does not work and the other one does not show that the authors want to indicate. By the way none of these refs are the ones shown in the manuscript (original ref. 68) and this ref.68 does not show what the authors refer either.
As said below, you cannot induce cell cycle progression by targeting the cell cycle machinery. This is a basic concept in Cell Biology, and a major defect in the manuscript (text and Figure 2), and it cannot be published carrying these errors.
The authors may consider reading several reviews on microRNAs and the cell cycle and the effect of several microRNAs on the G1 checkpoint (which is the contrary of cell cycle progression; see for instance https://www.ncbi.nlm.nih.gov/pmc/articles/PMC3740202/)
Again, as a general message, the beauty of the reviews is to critically evaluate previous data so that everything makes sense as a whole, and this section of the review is difficult to understand and believe as it is.
" Previous comment and answer:
g) Lines 205-208. “miR-302 promotes S-phase entry by direct targeting components of the cell cycle pathway, such as cyclin D1 and D2, CDK2, CDK4, etc.” This is again wrong. If you target these genes, you prevent S-phase entry
Thank you for the suggestion. However, the statement is generally true: miR-302 can promote S-phase entry by directly targeting components of cyclin D1 and D2, CDK2, and CDK4. It has been
reported that miR-302 play a role in cell cycle regulation, specifically in promoting the transition
from the G1 to S phase of the cell cycle. MiR-302 is known to directly target and suppress the
expression of cyclin D1, cyclin D2, CDK2, and CDK4, which are critical regulators of cell cycle
progression. By targeting these components, miR-302 inhibits their expression, leading to a
disruption in the regulation of the G1/S transition. This, in turn, promotes the entry of cells into the
S phase of the cell cycle (Doi: 10.1042/BJ20071635 and 10.1016/j.stem.2011.03.001).
However, we have accordingly modified the above sentences which can be found in lines 265-268:
In human ESCs, miR-302 modulates G1/S transition and consequently promotes S phase entry
by directly targeting components of the cell cycle pathway, such as CYCLIN D1, and D2, cyclin-
dependent kinase 2 (CDK2), CDK4, RB, E2F1, P130 and CDK6 [68].
Author Response
Dear Reviewer 3,
Thank you for reviewing our revised manuscript in detail again. We apologize for wrongly providing a reference and adding a non-appropriate references in our answer. Thank you for the suggested article (MiR-294/-302 promotes proliferation, suppresses the G1-S restriction point, and inhibits embryonic stem cell differentiation through separable mechanisms. Yangming Wang; Collin Melton; Ya-Pu Li; Archana Shenoy; Xin-Xin Zhang; Deepa Subramanyam; Robert Blelloch, Cell Rep. 2013 Jul 11; 4(1): 99–109.,doi:10.1016/j.celrep.2013.05.https://www.ncbi.nlm.nih.gov/pmc/articles/PMC3740202/). We included it in the references and in the manuscript text.
We have also modified Fig.2.
The corrected text of the manuscript and the description of Figure 2 are as follows:
Page 7, Line 267-269:
“… In addition to repressing CYCLIN D1/D2 and other cell cycle components, it promotes S phase entry through an alternative pathway [94] …. “
Page 8, Line 299-300:
Figure 2.
“… Additionally, they promote S phase entry via an alternative Rb-independent pathway….”
This manuscript is a resubmission of an earlier submission. The following is a list of the peer review reports and author responses from that submission.
Round 1
Reviewer 1 Report
This is an outstanding comprehensive literature article about specific miRNA clusters on the maintenance and regulating of embryonic development, pluripotent and self-renewal of stem cells in embryonic stem cell in vertebrates.
Although it is a very comprehensive review on an important field of study, the manuscript will benefit from changes regarding the readability and relevance.
Despite the kind of research is a review, the authors should describe a research gap that this manuscript will elaborate on. What is the importance of the subject that readers want to study this paper? And what is novelty?
Furthermore, it is not clear what the implementations are for stem cell-specific miRNA clusters from a clinical or pre-clinical point of view.
Next, although stem cell clusters are the main topic of the study, it should be more extensively described. Now an extensive amount of text is used to explain the miRNA biogenesis, which is not the main topic of the study. I would suggest to elaborate on the clusters.
It must be more clear that miRNAs from clusters are often expressed in the same manner but differ by seed sequence and therefore have different target genes, which lead to differences in biological functions.
It is not clear why this research focusses on human, mouse, rabbit and chicken. Is this the research gap?
See https://doi.org/10.1007/s12015-018-9808-y as review paper on the role of miRNA in cell-cycle regulation.
Author Response
"Please see the attachment."

Reviewer 2 Report
General comments:
- The headlines of sections have a lot of grammar errors and do not reflect the content described. Therefore, I would suggest the authors re-writing them.
- Many grammar errors, so moderate editing of the English language and style is required.
Section 3
- Line 61-62 the description of miRNA-mediated repression is not accurate. Please pay attention to the difference in miRNA-mediated gene regulation in plants and mammals.
- The definition of C19MC miRNA is missing.
Section 4
- There is no definition of miRNA cluster, and the authors’ description also puzzled readers sometimes. I would suggest the authors make a clear description between the miRNA family and miRNA cluster are different. Please see the article (DOI: 10.1007/978-1-4419-9863-7_1121) for details about the definition of miRNA cluster.
- How do the authors define miRNA clusters that are specific in stem cells? Are they only expressed in stem cells or they are just found to be functional in SCs? They mean the latter and this has to be clearly stated.
- The sub-headers are miss leading and should be something like miRNAs regulating stemness or differentiation of ESCs.
- Line 174-175 the author is writing about miRNA clusters but they described them as a miRNA family. This has to be corrected.
- Line 180-182 miR-302 is regulated by TF in both human and mouse cells, so they should use miR-302, not mir-302 which is a syntax for mouse miRNA. The same issue for lines 191-194.
- Line 188-189 here they are describing mouse miRNAs, so they should use mir-XXX.
Section 5
- The subtitle is a bit misleading. As there is no explanation why the miRNA cluster reviewed and discussed by the authors is more important than others? It is more appropriate if they change the title to “the most well-studied miRNA clusters in ESCs”.
- Figure 1.
· The miRNA cluster is not vertically aligned making the comparison difficult. In addition, for the 5p strand miR-367 and miR-302’s seed regions are different from others while for the 3p strand, only the miR-367’s seed region is different from the rest cluster members.
· Why the authors showed only the conservation in four species, how about the other ones such as zebra and fruit fly for which miRNAs are also well documented and studied?
· The visualization of the figure should be reconsidered. As the chromosome bar did not deliver any relevant information described by the authors. The authors may consider removing it and reorganizing the sequences in a logical and reasonable manner.
- Section 5.1.2
· What does dynamic expression profile mean? Does the expression of the miRNA cluster changes over time?
· The authors just describe the observation that some of the cluster members are expressed at different levels during the development of embryos. But I did not see any discussion on the function of these miRNAs. What is the biological implication for the quick change of the expression of the miRNA cluster in embryonic stages?
- Section 5.1.3. I was left wondering how gga-miR-302b promotes the proliferation and apoptosis of PGCs.
- Section 5.1.4. This section is the best-written part, as the authors clearly explained the targets of miRNAs and their biological outputs of the corresponding gene regulation by the miRNA cluster. Such a strategy should be applied to the whole manuscript. However, there are quite some grammar errors that need to be corrected.
- Figure 2. The authors should add a brief figure caption to explain the content of the figure.
- Section 5.2
· What is the host gene of the C19MC cluster?
· Line 294-295. How 46 pre-miRNAs produce only 56 mature miRNAs. The number of mature miRNAs should be 92, right?
· Figure 3. The seed region in the 5p strand is not highlighted. Why is miR-1323 highlighted in red? The label of 5’ and 3’ on DNA are not correct.
· Line 348, how do researchers identify the origin of the circulating miRNA cluster from the tumor or normal cells?
· Section 5.2.3. What algorithms have been used for predicting miRNA targets? There are many papers studying miRNA’s role in regulating EMT in circuits such as feedback loops but were not discussed at all.
- Section 5.3
· Figure 4. It will definitely be interesting to show the genomic coordination of miRNAs in the host genes. In addition, if you want to show the h
· Figure 5. The title should be miR-209/295. What does the arrow with a slash between Dkk-1 and Wnt mean? A brief description of the figure is needed.
Section 6
The table does not do the job of making conclusions and a summary of the study without a proper description in the main text.
In addition, I have not seen any opinions or discussions from the authors’ side. This is supposed to be the most important part of a review paper because it can inform the readers or new researchers about the current development of miRNA research in stem cells and also point out the future direction of the field.
## Typos and grammar
As there are many typos and grammar errors I just listed some identified in the first 2 sections
Line 50 should be 3’ end
Line 88-90 missing references on the specified clusters in ESCs.
Line 143 should be miR-155 and a redundant space should be removed.
Author Response
"Please see the attachment."

Round 2
Reviewer 1 Report
The authors have not properly answered my comments. In my opinion, the manuscript has not been sufficiently improved. For example, I asked about the clinical relevance, the clinical implications and the research gap. Authors responded with: acquiring knowledge. I don't think the manuscript is novel enough. The English language used also needs a lot of improvement.
Reviewer 2 Report
NA